# Generating and Checking DNN Verification Proofs

**Hai Duong**
George Mason University
Fairfax, VA 22030
hduong22@gmu.edu

**ThanhVu Nguyen**
George Mason University
Fairfax, VA 22030
tvn@gmu.edu

**Matthew B. Dwyer**
University of Virginia
Charlottesville, VA 22904
matthewbdwyer@virginia.edu

## Abstract

Deep Neural Networks (DNN) have emerged as an effective approach to implementing challenging subproblems. They are increasingly being used as components in critical transportation, medical, and military systems. However, like human-written software, DNNs may have flaws that can lead to unsafe system performance. To confidently deploy DNNs in such systems, strong evidence is needed that they do not contain such flaws. This has led researchers to explore the adaptation and customization of software verification approaches to the problem of neural network verification (NNV). Many dozens of NNV tools have been developed in recent years and as a field these techniques have matured to the point where realistic networks can be analyzed to detect flaws and to prove conformance with specifications. NNV tools are highly-engineered and complex may harbor flaws that cause them to produce unsound results.

We identify commonalities in algorithmic approaches taken by NNV tools to define a verifier independent proof format—activation pattern tree proofs (APTP)—and design an algorithm for checking those proofs that is proven correct and optimized to enable scalable checking. We demonstrate that existing verifiers can efficiently generate APTP proofs, and that an `APTPchecker` significantly outperforms prior work on a benchmark of 16 neural networks and 400 NNV problems, and that it is robust to variation in APTP proof structure arising from different NNV tools. `APTPchecker` is available at: https://github.com/dynaroars/APTPchecker.

## 1 Introduction

As deep neural networks (DNNs) become integral components of critical systems such as autonomous vehicles [1], medical decision-making [2], and robotics [3], it is imperative to rigorously verify their behavior. In recent years, the research community has developed a wide-range of algorithmic techniques to verify DNN properties and incorporated them into tools that now scale to realistic DNN models millions of neurons [4]. These advances have enabled verification of properties such as robustness to input perturbations and conformance to safety specifications [4, 5, 6, 7, 8].

However, despite the progress in algorithmic advances, a fundamental question remains: *"How can we trust the results produced by DNN verification tools?"* While existing tools emit counterexamples when properties are violated (i.e., SAT results), there is no mechanism to independently validate results when properties are proven to hold (i.e., UNSAT claims). Recent competitions such as VNN-COMP [4] have revealed correctness issues in multiple tools, including cases where a verifier incorrectly declared a property to be proven even when a counterexample exists. These errors are difficult to detect and debug due to the complexity of verifier implementations, which often exceed tens of thousands of lines of code and employ intricate optimization techniques, e.g., top of the line DNN verification tools such as $\alpha\beta$-`CROWN` [7] and `NeuralSAT` [9] have 20k SLOC implementations with complex algorithms that may harbor bugs. Without a mechanism to independently validate

39th Conference on Neural Information Processing Systems (NeurIPS 2025).

verification results, correctness of DNN verification tools cannot be assured and therefore posing a serious obstacle to deploying DNNs in safety-critical domains.

To address this, we propose *proof-producing DNN verification*: an approach in which verifiers emit a formal proof object that encodes the reasoning steps behind the verification result, and a separate, minimal proof checker certifies the proof's validity. This paradigm, long established in classical logic and SAT solving [10, 11, 12], brings transparency, auditability, and trust to the verification process.

More specifically, we analyze the broad class of Branch-and-Bound (BaB) DNN verification algorithms and reveal that they share two commonalities: (1) they refine the abstractions they use by performing case *splitting* to reason about the different phases of neuron activation, and (2) within cases they perform reasoning steps that can be formulated within the broad class of *mixed integer linear programming* (MILP) problems. Based on these insights, we show that BaB DNN verification naturally emit *activation pattern tree proofs* (APTP), which are a compact representation of the reasoning steps performed by the verifier (§3.1). We also define a verifier independent APTP format that can be efficiently generated on-the-fly during DNN verification (§3.2). Finally, we resent the APTPchecker algorithm along with a suite of optimizations and implement an independent APTPchecker prototype tool that has a small-footprint (800 SLOC) and validates APTP proofs using standard MILP solving.

This paper makes the following contributions:

- We identify commonalities in BaB DNN verification algorithms and show how they can be minimally extended to generate proofs of unsatisfiability (§3.1).
- We define a verifier-independent, compact, and SMTLib [13]-based human-readable proof format, APTP, that captures the reasoning steps of BaB verifiers (§3.2).
- We implement the APTPchecker tool to independently and efficiently check APTP proofs (§4).
- We evaluate our work on a benchmark of 400 verification problems involving 16 networks, including large models (up to 1.7M parameters) (§5), and demonstrate that APTP and APTPchecker are robust to variation in proof structure arising from different DNN verification algorithms.

It is important to note that our goal is *not* to create a new DNN verifier, but to verify the correctness of results produced by *existing verifiers*. Similar to SAT/SMT proof checking, where verifiers produce proofs that are independently checked (e.g., DRAT proofs [10]), we propose such an approach for DNN verification. We envision APTP and APTPchecker as a step toward *DNN verifier accountability*, standardizes proof formats, supports independent checking, and can be integrated into future VNN-COMP iterations to strengthen trust in verification results.

## 2 Background

**Deep Neural Network**   A *neural network* consists of an input layer, multiple hidden layers, and an output layer. The output of a DNN is obtained by progressively computing the values of neurons in each layer. Specifically, the value of a hidden neuron $y$ is $ReLU(\sum_i^n w_i v_i + b)$, where $b$ is the bias, $w$s are the weights of $y$, $v$s are the neurons of preceding layer, $\sum_i^n w_i v_i + b$ is the *affine transformation*, and $ReLU(x) = \max(x, 0)$ is the *activation function*. The Rectified Linear Unit (ReLU) is a representative of a broad class of *piece-wise linear* activations that could be supported by our approach. A ReLU neuron is *active* if its input value is greater than zero and *inactive* otherwise.

**DNN Verification**   Given a DNN $\mathcal{N}$ and a property $\phi$, the *DNN verification problem* asks if $\phi$ is a valid property of $\mathcal{N}$. In modern DNN verification, $\phi(x, y) := \phi_{in}(x) \Rightarrow \phi_{out}(y)$, where $\phi_{in}$ is a property over the inputs and $\phi_{out}$ is a property over the outputs of $\mathcal{N}$. This form of properties has been used to encode safety and security requirements of DNNs [14, 15].

DNN verification then can be formulated as checking the satisfiability of:

$$\alpha \wedge \phi_{in} \wedge \neg\phi_{out} \tag{1}$$

where $\alpha$ is the encoding of $\mathcal{N}$. A DNN verifier attempts to find a *counterexample* input to $\mathcal{N}$ that satisfies $\phi_{in}$ but violates $\phi_{out}$. If Eq. 1 is unsatisfiable (e.g., no such counterexample exists), $\phi$ is a *valid* property of $\mathcal{N}$ and *invalid* otherwise.

**Alg. 1.** The BaB$_{\text{NV}}$ algorithm with proof generation.

**input** : DNN $\mathcal{N}$, property $\phi_{in} \Rightarrow \phi_{out}$
**output** : $(unsat, proof)$ if property is valid, otherwise $(sat, cex)$

1  $ActPatterns \leftarrow \{\emptyset\}$ // initialize verification problems
2  $proof \leftarrow \{\ \}$ // initialize proof tree
3  **while** $ActPatterns$ **do** // main loop
4      $\sigma_i \leftarrow \text{Select}(ActPatterns)$ // process problem $i$-th
5      **if** $\text{Deduce}(\mathcal{N}, \phi_{in}, \phi_{out}, \sigma_i)$ **then**
6          $(cex, v_i) \leftarrow \text{Decide}(\mathcal{N}, \phi_{in}, \phi_{out}, \sigma_i)$
7          **if** $cex$ **then return** $(sat, cex)$ // found a valid counter-example
8          $ActPatterns \leftarrow ActPatterns \cup \{\sigma_i \wedge v_i\ ;\ \sigma_i \wedge \overline{v_i}\}$ // new activation patterns
9      **else** // detect a conflict
10         $proof \leftarrow proof \cup \{\sigma_i\}$ // build proof tree
11 **return** $(unsat, proof)$

For the widely-used ReLU activation problem, this problem becomes a search for *activation patterns*, i.e., boolean assignments representing activation status of neurons, that lead to satisfaction the formula in Eq. 1. Modern DNN verification techniques [7, 6, 9, 16, 17, 18] all adopt this idea and search for satisfying assignments.

**Related Work (more details in Apdx. D)**   Proof checking is a well-established area in constraint solving, particularly in SAT/SMT solving, with significant work on clausal proof generation and verification, such as DRAT for SAT solvers and various proof checkers like DRAT-trim and LRAT [10, 19, 12]. SMT solvers, such as Z3 and veriT, also produce proofs that can be reconstructed in proof assistants, and other solvers like MathSAT5, SMTInterpol, and CVC5 have similar capabilities [20, 21, 22, 23, 24]. However, DNN verification is a newer field, with limited research on proof checkers.

The only existing proof checking work for DNNs focuses on the `Marabou` DNN verification tool [25], using Farkas's lemma and implemented in the Imandra framework [26]. In contrast, we introduces a more expressive proof format, `APTP`, and stronger proof checker, `APTPchecker`, specifically designed for branch-and-bound verification (§3.1).

## 3  Proof Generation for DNN Verification

Major DNN verification approaches including $\alpha\beta$-`CROWN` [7], `NeuralSAT` [16], `PyRAT` [6], `nnenum` [17], and `Marabou` [27] all share a common "branch and bound" (BaB) search structure: (i) (branch) split into smaller subproblems by using *neuron splitting*, which decides boolean values representing neuron activation status, and (ii) (bound) use abstraction and LP solving to approximate bounds on neuron values to determine the satisfiability of the partial activation pattern formed by the split. We leverage this commonality to bring proof generation capabilities with minimal overhead to existing DNN verification tools.

In this paper we focus on checking proofs of unsatisfiability (unsat). A counterexample, $c$, returned by a verifier is an input that is purported to violate the property. This constitutes a proof of satisfiability (sat) and can easily be checked by evaluating $\phi(c, N(c))$. In contrast, unsat proof, which explains why *no possible inputs* can violate the property, is inherently more complex to generate (§3), requires a more sophisticated encoding (§3.2), and an efficient checking algorithm (§4).

### 3.1  Branch-and-Bound DNN Verification

Alg. 1 illustrates BaB$_{\text{NV}}$, a reference architecture [28] for modern DNN verifiers based on the branch-and-bound (BaB) framework. BaB$_{\text{NV}}$ takes as input a ReLU-based DNN and a property of interest. It iteratively alternates between two core components: Decide (line 6), which performs neuron-splitting by assigning an activation status (active/inactive) to a neuron, and Deduce (line 5), which checks the feasibility of the current activation pattern and prunes infeasible branches.

Our key insight is that the BaB architecture of BaB$_{\text{NV}}$ naturally supports proof generation. To realize this, we augment BaB$_{\text{NV}}$ with a proof tree structure, stored in the $proof$ variable (line 2). We also

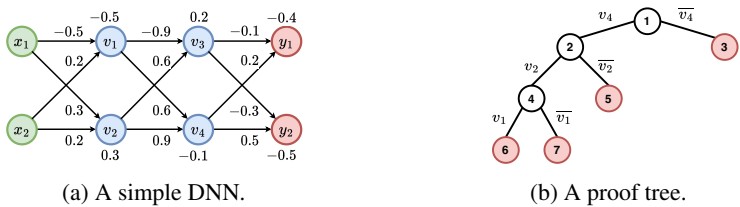

(a) A simple DNN.

(b) A proof tree.

Fig. 1: Example of verifying $(x_1, x_2) \in [-2.0, 2.0] \times [-1.0, 1, 0] \Rightarrow (y_1 > y_2)$.

instrument $\text{BaB}_{\text{NV}}$ so that each branching decision made during the Decide step is explicitly recorded into this tree (line 10). Each node in the binary proof tree represents a neuron, and its left and right children correspond to the two possible activation decisions (active or inactive).

**Example** Fig. 1a illustrates a DNN and how $\text{BaB}_{\text{NV}}$ determines unsatisfiability (i.e., verifies the problem) and generates the unsat proof in Fig. 1b. First, $\text{BaB}_{\text{NV}}$ initializes the activation pattern set $ActPatterns$ with an empty activation pattern $\emptyset$. Then $\text{BaB}_{\text{NV}}$ enters a loop (line 3-line 10) to search for a satisfying assignment or a proof of unsatisfiability.

**1st iteration**: $\text{BaB}_{\text{NV}}$ selects the only available activation pattern $\emptyset \in ActPatterns$, and calls Deduce to check the feasibility of the problem based on the current activation pattern. Deduce uses abstraction to approximate that from the input constraints the output values are feasible for the given network. Since Deduce cannot determine infeasibility, $\text{BaB}_{\text{NV}}$ invokes Decide to randomly select a neuron to split. Suppose it selects neuron $v_4$, which results in the original problem being divided into two independent subproblems: one where $v_4$ is active, and another where $v_4$ is inactive. $\text{BaB}_{\text{NV}}$ then adds $v_4$ and $\overline{v_4}$ to $ActPatterns$.

**2nd iteration**: $\text{BaB}_{\text{NV}}$ has two subproblems that can be processed in parallel. For the first subproblem with $v_4$, Deduce cannot decide infeasibility, so it selects $v_2$ to split. It then conjoins $v_4$ with $v_2$ and then with $\overline{v_2}$ and adds both conjuncts to $ActPatterns$. For the second subproblem with $v_4$ inactive (i.e., $\overline{v_4}$), Deduce determines that the problem is unsatisfiable and $\text{BaB}_{\text{NV}}$ saves $\overline{v_4}$ to the proof tree, as node 3, to indicate one unsatisfiable pattern, i.e., whenever the network has $v_4$ being inactive, the problem is unsatisfiable.

**3rd iteration**: $\text{BaB}_{\text{NV}}$ has two subproblems for $v_4 \wedge v_2$ and $v_4 \wedge \overline{v_2}$. For the first subproblem, Deduce cannot decide infeasibility, so it selects $v_1$ to split. It then conjoins $v_1$ and then $\overline{v_1}$ to the current activation pattern and adds them to $ActPatterns$. For the second one, Deduce determines that the problem is unsatisfiable and $\text{BaB}_{\text{NV}}$ saves the $v_4 \wedge \overline{v_2}$ to the proof tree, as node 5.

**4th iteration**: $\text{BaB}_{\text{NV}}$ has two subproblems for $v_4 \wedge v_2 \wedge v_1$ and $v_4 \wedge v_2 \wedge \overline{v_1}$. Both subproblems are determined to be unsatisfiable, and $\text{BaB}_{\text{NV}}$ saves them to the proof tree as nodes 6 and 7, respectively.

Finally, $\text{BaB}_{\text{NV}}$ has an empty $ActPatterns$, stops the search, and returns $unsat$ and the proof tree.

**The APTP proof tree** The resulting proof tree has a specific structure. First, it is a binary tree where each parent node must have children for both activation status values of a neuron. Second, it is a *proof tree* that captures unsatisfiability reasoning, i.e., each leaf holds the constraint showing the activation pattern encoded from the root to this leaf results in unsatisfiability. The tree in Fig. 1b demonstrates this structure. Each white node corresponds to a branching node where $\text{BaB}_{\text{NV}}$ makes decisions to split neurons. The red leaves correspond to the unsatisfiable patterns that are saved to the proof tree. Note that a leaf node implies the unsatisfiability of the subtree rooted at the leaf, e.g., node 3 encodes the unsatisfiability of a set of 8 activation patterns.

We leverage this structure to store the proof in the APTP format (§3.2) and to check it using the APTPchecker algorithm (§4).

## 3.2 The APTP Proof Language

We have shown in §3 that the broad class of $\text{BaB}_{\text{NV}}$ DNN verification techniques can generate a binary tree that represents a proof of unsatisfiability (§3). We define a standard proof format for specifying DNN proofs, APTP, that is human-readable, compact, and efficiently generated by verification tools

$$\begin{array}{ll}
\langle proof \rangle &::= \langle declarations \rangle \ \langle assertions \rangle \\
\langle declarations \rangle &::= \langle declaration \rangle \mid \langle declaration \rangle \ \langle declarations \rangle \\
\langle declaration \rangle &::= (\textbf{declare-const} \ \langle input\text{-}vars \rangle \ \textbf{Real}) \\
& \quad \mid (\textbf{declare-const} \ \langle output\text{-}vars \rangle \ \textbf{Real}) \\
& \quad \mid (\textbf{declare-pwl} \ \langle hidden\text{-}vars \rangle \ \langle activation \rangle) \\
\langle input\text{-}vars \rangle &::= \langle input\text{-}var \rangle \mid \langle input\text{-}var \rangle \ \langle input\text{-}vars \rangle \\
\langle output\text{-}vars \rangle &::= \langle output\text{-}var \rangle \mid \langle output\text{-}var \rangle \ \langle output\text{-}vars \rangle \\
\langle hidden\text{-}vars \rangle &::= \langle hidden\text{-}var \rangle \mid \langle hidden\text{-}var \rangle \ \langle hidden\text{-}vars \rangle \\
\langle activation \rangle &::= \text{ReLU} \mid \text{Leaky ReLU} \mid \ldots \\
\langle assertions \rangle &::= \langle assertion \rangle \mid \langle assertion \rangle \ \langle assertions \rangle \\
\langle assertion \rangle &::= (\textbf{assert} \ \langle formula \rangle) \\
\langle formula \rangle &::= (\langle operator \rangle \ \langle term \rangle \ \langle term \rangle) \\
& \quad \mid (\textbf{and} \ \langle formula \rangle +) \mid (\textbf{or} \ \langle formula \rangle +) \\
\langle term \rangle &::= \langle input\text{-}var \rangle \mid \langle output\text{-}var \rangle \\
& \quad \mid \langle hidden\text{-}var \rangle \mid \langle constant \rangle \\
\langle operator \rangle &::= < \mid \leq \mid > \mid \geq \\
\langle input\text{-}var \rangle &::= \text{X\_} \langle constant \rangle \\
\langle output\text{-}var \rangle &::= \text{Y\_} \langle constant \rangle \\
\langle hidden\text{-}var \rangle &::= \text{N\_} \langle constant \rangle \\
\langle constant \rangle &::= \textbf{Int} \mid \textbf{Real}
\end{array}$$

```
1  ; Declare variables
2  (declare-const X_0 Real)
3  (declare-const X_1 Real)
4  (declare-const Y_0 Real)
5  (declare-const Y_1 Real)
6  (declare-pwl N_1 ReLU)
7  (declare-pwl N_2 ReLU)
8  (declare-pwl N_3 ReLU)
9  (declare-pwl N_4 ReLU)
10 ; Input constraints
11 (assert (>= X_0 -2.0))
12 (assert (<= X_0  2.0))
13 (assert (>= X_1 -1.0))
14 (assert (<= X_1  1.0))
15 ; Output constraints
16 (assert (<= Y_0 Y_1))
17 ; Hidden constraints
18 (assert (or
19   (and (<  N_4 0))
20   (and (<  N_2 0)
21        (>= N_4 0))
22   (and (>= N_2 0)
23        (>= N_1 0)
24        (>= N_4 0))
25   (and (>= N_2 0)
26        (<  N_1 0)
27        (>= N_4 0))))
```

(a) The APTP proof language.                 (b) APTP example.

Fig. 2: The APTP format.

and processed by proof checkers. APTP is inspired by the SMTLIB format [13] used for SMT solving, which has also been adopted by the VNNLIB language [29] to specify DNN verification problems.

Fig. 2a presents the syntax of the APTP proof language. A proof consists of *declarations* and *assertions*. Declarations define input/output variables (real numbers) and hidden variables (with PWL activations like ReLU). Assertions encode preconditions over inputs and postconditions over outputs using logical formulas with comparisons and Boolean operators like and and or. More details on the syntax and semantics of APTP are available in (Apdx. A).

**Example** The APTP proof in Fig. 2b corresponds to the proof tree in Fig. 1b. The statement (and (< N_4 0)) corresponds to the rightmost path of the tree with $\overline{v_4}$ decision (leaf 3). The statement (and (< N_2 0) (>= N_4 0)) corresponds to the path with $v_4 \land \overline{v_2}$ (leaf 5).

The APTP language is intentionally designed to (a) omit explicit weights and biases to reduce the size of the proof structure, and (b) explicitly encode a DNF structure to enable easy parallelization. The weights and biases of the DNN are already recorded in the ONNX format [30], which serves as a standard input to both verification tools and APTP checkers, like the one we describe in §4.

Note that the APTP language can be extended to support other piece-wise linear activation functions, e.g., Leaky ReLU. Particularly, $Leaky(x, a) = x$ if $x \geq 0$ else $ax$ are expressible by APTP, e.g., (>= Ni 0) for on and (< Ni 0) for off, which is essentially identical to ReLU, while the MILP encoding described in §4.1 would handle the semantics of Leaky operations.

### 3.3 Handling Input Splitting

For networks with a small number of inputs (e.g., $\leq 50$), some verifiers [31, 32, 7, 16] employ *input-splitting* strategies that partition the input domain rather than branching on neuron activations as in BaB. Our APTP proof generation and format can naturally represent these cases without modification. More specifically, each input split can be encoded as a conjunction of input-range constraints describing the corresponding subspace, replacing the "hidden activation" constraints used in neuron splitting. For example, splitting on $x_1$ at 0 yields two subproblems, each defined by intervals on $x_1$ and $x_2$, as shown below.

```
1 (assert (or
2   (and (>= X_0 -2.0) (<= X_0  2.0) (>= X_1  0.0) (<= X_1  1.0))
3   (and (>= X_0 -2.0) (<= X_0  2.0) (>= X_1 -1.0) (<= X_1  0.0))
4 ))
```

**Alg. 2.** `APTPchecker` algorithm.

---

**input** : DNN $\mathcal{N}$, property $\phi_{in} \Rightarrow \phi_{out}$, $proof$
**output** : $certified$ if proof is valid, otherwise $uncertified$

---

1 **if** $\neg$ `RepOK` $(proof)$ **then** `RaiseError`(*Invalid proof tree*)
2 $model \leftarrow$ `CreateStabilizedMILP`$(\mathcal{N}, \phi_{in}, \phi_{out})$ // initialize MILP model with inputs
3 **while** $proof$ **do**
4     $node \leftarrow$ `Select`$(proof)$ // get node to check
5     $model \leftarrow$ `AddConstrs`$(model, node)$ // add corresponding constraints
6     **if** `CheckFeasibility`$(model)$ **then**
7        **return** $uncertified$ // cannot certify

8 **return** $certified$

---

In this setting, the proof generation step emits input-subspace constraints instead of hidden-layer constraints, and the proof tree structure remains unchanged. Essentially, we treat each input dimension as a "neuron" and split on its range, similar to neuron splitting in BaB.

## 4 Checking `APTP` Proofs

After generating proofs, the next step is to validate them. The goal is to separate proof generation from proof checking, where existing verifiers generate UNSAT proofs while a separate, independent checker independently validates these proofs. To this end, we introduce a proof checker, `APTPchecker`, that validates APTP proofs. First, `APTPchecker` eschews all optimizations and complexities in modern verifiers. It only needs to validate the final activation pattern claims, regardless of the sophisticated search strategies, bound tightening techniques, or pruning heuristics used by the verifiers. Next, `APTPchecker` achieves a substantially reduced trusted code base, requiring only 800 SLOC compared to 20K SLOC in verifiers, while providing verifier independence where the same proof format works across multiple verification tools. Finally, `APTPchecker` also uses the MILP solver as a black box, allowing multiple solvers to be employed to increase confidence in results.

### 4.1 The Core `APTPchecker` Algorithm

The goal of `APTPchecker` is to verify that the `APTP` tree generated by a DNN verification tool is correct (i.e., the proof tree is a proof of unsatisfiability of the DNN verification problem). `APTPchecker` thus must verify that the constraint represented by each *leaf* node in the proof tree is unsatisfiable. To check each node, `APTPchecker` forms an MILP problem consisting of the constraint in Eq. 1 (the DNN, the input condition, and the negation of the output) with the constraints representing the activation pattern encoded by the tree path to the leaf node. `APTPchecker` then invokes an LP solver to check that the MILP problem is infeasible, which indicates unsatisfiability of the leaf node.

**Core Algorithm** Alg. 2 shows a minimal (core) `APTPchecker` algorithm, which takes as input a DNN $\mathcal{N}$, a property $\phi_{in} \Rightarrow \phi_{out}$, a proof tree $proof$, and returns $certified$ if the proof tree is valid and $uncertified$ otherwise. `APTPchecker` first checks the validity of the proof tree (line 2), i.e., the input must represent a proper APTP proof tree (§3.2). If the proof tree is invalid, `APTPchecker` raises an error. `APTPchecker` next creates a MILP model (line 2) representing the input. `APTPchecker` then enters a loop (line 3) that selects a (random) leaf node from the proof tree (line 4) and adds its MILP constraint to the model (line 5). It then checks the model using an LP solver to determine whether the leaf node is unsatisfiable. If the LP solver returns feasibility, `APTPchecker` returns $uncertified$, i.e., it cannot verify the input proof tree. `APTPchecker` continues until all leaf nodes are checked and returns $certified$, indicating the proof tree is valid.

**Example** For the `APTP` proof in Fig. 2b, we need to check that the four leaf nodes 3, 5, 6, and 7 of the proof tree in Fig. 1b are unsatisfiable. Assume `APTPchecker` first selects node 3, it forms the MILP problem for leaf node 3 by conjoining the constraint representing $0.6v_1 + 0.9v_2 - 0.1 \leq 0$ (i.e., $\overline{v_4}$) with the constraints in Eq. 1 representing the input ranges and the DNN with the objective of optimizing the output. `APTPchecker` then invokes an LP solver, which determines that this MILP is infeasible, i.e., leaf node 3 indeed leads to unsatisfiability. `APTPchecker` continues this process for the other three leaf nodes and returns $certified$ as all leaf nodes are unsatisfiable.

**MILP Formulation**

`APTPchecker` formulates MILP problems [33] and checks for feasible solutions using off-the-shelf LP solving. Formally, the MILP problem is defined as:

$$
\begin{aligned}
&\text{(a)} \quad z^{(i)} = W^{(i)}\hat{z}^{(i-i)} + b^{(i)}; &&\text{(b)} \quad y = z^{(L)}; x = \hat{z}^{(0)};\\
&\text{(c)} \quad \hat{z}_j^{(i)} \geq z_j^{(i)}; \hat{z}_j^{(i)} \geq 0; &&\text{(d)} \quad a_j^{(i)} \in \{0,1\}; &&&(2)\\
&\text{(e)} \quad \hat{z}_j^{(i)} \leq a_j^{(i)} u_j^{(i)}; \hat{z}_j^{(i)} \leq z_j^{(i)} - l_j^{(i)}(1 - a_j^{(i)});
\end{aligned}
$$

where $x$ is input, $y$ is output, and $z^{(i)}$, $\hat{z}^{(i)}$, $W^{(i)}$, and $b^{(i)}$ are the pre-activation, post-activation, weight, and bias vectors for layer $i$, respectively. This encodes precisely the semantics of a ReLU-based DNN: (a) the affine transformation computing the pre-activation value for a neuron; (b) the inputs and outputs in the DNN; (c) assertion that post-activation values are non-negative and no less than pre-activation values; (d) neuron activation status indicator variables that are either 0 or 1; and (e) constraints on the upper, $u_j^{(i)}$, and lower, $l_j^{(i)}$, bounds of the pre-activation value of the $j$th neuron in the $i$th layer. Deactivating a neuron, $a_j^{(i)} = 0$, simplifies the first of the (e) to $\hat{z}_j^{(i)} \leq 0$, and activating a neuron simplifies the second to $\hat{z}_j^{(i)} \leq z_j^{(i)}$, which is consistent with $\hat{z}_j^{(i)} = max(z_j^{(i)}, 0)$.

**Correctness** Alg. 2 returns $certified$ iff the input APTP proof tree is unsatisfiable. This proof tree encodes a disjunction of constraints, one per tree path, where each constraint represents an activation pattern of the network (the leaf node). Then each problem is reduced to a simple LP that exactly captures the semantics of the DNN for a specific activation pattern and thus, the algorithm introduces no approximations, i.e., it is sound and complete.

Note that this correctness argument assumes that the LP solver is correct. In practice multiple solvers could be used to guard against errors in that component. It is standard for proof checkers to assume the correctness of a small set of external tools, e.g., checkers that use theorem provers assume the correctness of the underlying prover [34].

**Implementation and Validation** `APTPchecker` is written in Python, and uses Gurobi [35] for LP solving. The core `APTPchecker` algorithm (Alg. 2) consists of 600 SLOC, while optimizations use an additional 200 SLOC. Currently, `APTPchecker` supports ReLU-based feed-forward (FNNs) and convolutional neural networks (CNNs). `APTPchecker` uses ONNX for neural networks and outputs APTP proofs. In addition, we used the CrossHair [36] symbolic execution tool to check the correctness of the core algorithm in `APTPchecker`. Specifically, CrossHair confirmed that key postconditions hold, e.g., that `APTPchecker` returns $certified$ if and only if all leaf nodes in the proof tree are formally proven. While the verification is not exhaustive (CrossHair only explore program paths up to a certain depth), this increases confidence in the implementation's correctness up to certain depth. A detailed discussion is provided in Apdx. B.

## 4.2 Optimizations

Our `APTPchecker` implementation employs several optimizations to improve efficiency, especially for large proof trees. It uses *neuron stabilization* to identify stable neurons (either active or inactive) and replace disjunctive constraints with linear ones, and simplifying the MILP problem and reducing the work of the LP solver. Additionally, it employs *pruning of leaf nodes* and backtracking to check parent nodes only when necessary, reducing the number of LP problems to be solved. Finally, `APTPchecker` leverages the tree structure of APTP proof to *parallelize* the checking of leaf nodes, making the verification process scale better to large proof trees. Additional details on these optimizations are available in Apdx. C.

## 5 Evaluation

We evaluate our work using the following research questions: **RQ1** (§5.1): How does proof generation and checking perform with existing verifiers? **RQ2** (§5.2): What factors impact proof checking performance? **RQ3** (§5.3): How does this work compare to prior work? **RQ4** (§5.4): What are some unsound cases that we can detect?

Tab. 1: Benchmarks consist of a 8 neural networks comprised of varying numbers of CNN (C) and FNN (F) layers, neurons, and parameters, each paired with 25 properties to form UNSAT verification instances.

| Name | Num. | Layers | Neurons | Param. | Instances Num. |
|------|------|--------|---------|--------|----------------|
| | | **Networks** | | | |
| CNN | 8 | 1-2C;1F | 320-3920 | 41K-180K | 200 |
| FNN | 8 | 2-6F | 64-3072 | 27K-1.7M | 200 |

Tab. 2: `APTPchecker` performances in seconds (mean/median/max).

| Verifier | FNN | | CNN | |
|----------|-----|--|-----|--|
| | Verify | Check Proof | Verify | Check Proof |
| $\alpha\beta$-CROWN | 4/9/481 | 3/19/870 | 6/16/50 | 13/67/866 |
| NeuralSAT | 9/16/304 | 2/10/599 | 12/30/64 | 7/42/825 |

**Benchmarks**  We evaluate on UNSAT verification problems selected from the benchmark suite introduced in [9], which includes ACAS Xu, RESNET_A/B, CIFAR2020, MNISTFC, and MNIST_GDVB. Specifically, MNISTFC contains networks with 2, 4, and 6 layers, each with 256 ReLUs per layer. MNIST_GDVB networks, generated using GDVB [37], range from 2-6 layers with 16-512 neurons per layer, yielding 64-3072 ReLU counts total. As with prior work [5] we exclude ACAS Xu, which has networks with low input dimensions and did not even need to use BaB on activation space to be solved. We also exclude RESNET, which are currently not supported by the `APTPchecker`. Note that this a straightforward engineering limitation and there is no fundamental reason the checking algorithm is not applicable.

From CIFAR2020, we selected CNN models and varied convolutional sizes and depths; from MNISTFC and MNIST_GDVB, we chose 8 FNNs of diverse sizes. For robustness properties, we use local perturbation radii ranging from 0.01 to 0.09 in $L_\infty$ norm. For each network, we randomly sampled local robustness properties until we obtained 25 UNSAT instances, yielding 200 CNN and 200 FNN problems (400 total) as shown in Tab. 1.

**Baselines**  We adapted two verifiers: $\alpha\beta$-CROWN and NeuralSAT, to generate APTP proofs. We also compare our work with the Marabou verifier and its proof checker [38].

**Metrics**  To assess performance we use the two common metrics in the verification community [4]: (i) time to solve and (ii) number of problems solved. We record time to verify, generate, and check proofs, using a 1000-second timeout. A problem is "solved" if all steps complete within time limit.

**Setup**  All experiments were run on a Linux machine with an AMD Threadripper 64-core 4.2GHZ CPU, 128GB RAM, and an NVIDIA GeForce RTX 4090 GPU with 24 GB VRAM.

### 5.1  RQ1 : Proof Generation and Checking

Fig. 3 shows cactus plots for proof generation and checking with the underlying NeuralSAT and $\alpha\beta$-CROWN verifiers. In cactus plots like this, lines that extend further on the x-axis are better – more problems solved – and lines that are lower are better – faster solve times. The dashed lines show the performance of the verifier and the solid lines show the performance of the verifier, proof generation, and the proof checker.

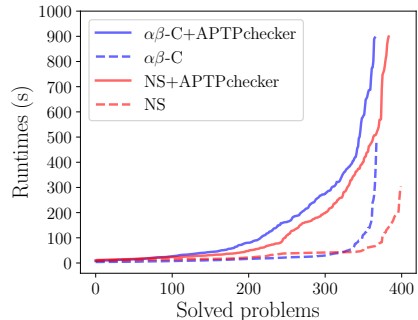

Fig. 3: Runtimes and problems solved.

Fig. 3, with additional timing details given in Tab. 2, show that checking proofs is *slower* than verification itself. Modern verification tools rely on sophisticated abstractions and optimizations (e.g., $\alpha\beta$-CROWN and NeuralSAT employ advanced bound propagation, pruning heuristics, and leverage GPU acceleration), but these enhancements are precisely what make verification complex and error-prone. In contrast, APTPchecker

Tab. 3: subproofs vs. MILP complexity.

| Verifier | Num. subproofs | | MILP Complexity | |
|---|---|---|---|---|
| | Mean | Median | Mean | Median |
| $\alpha\beta$-CROWN | 230 | 180 | 414 | 179 |
| NeuralSAT | 95 | 36 | 601 | 545 |

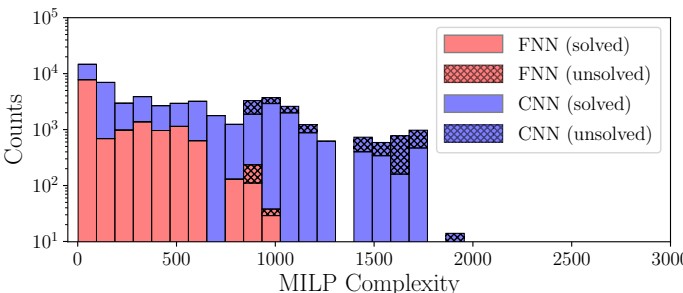 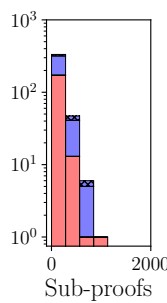

Fig. 4: Number of constraints and subproofs per problem.

deliberately eschews all verifier-specific optimizations and systematically validates each proof node using MILP (e.g., we use Gurobi which runs on CPU but can be replaced with other solvers), prioritizing reliability over speed.

Overall, both verifiers were able to generate proofs for all problems with APTP, and APTPchecker is able to check between 93.7% and 99.4% of the proofs that are generated. This demonstrates that the APTP is able to encode proofs generated by differing neural network verification algorithms, and that APTPchecker can check them.

### 5.2 RQ2 : Proof Checking Analysis

Tab. 3 shows statistics on the number of *subproofs*—the number of MILP calls made by APTPchecker (equivalent to the number of leaf nodes in the proof tree if no pruning is used), and the *MILP complexity*—the number of constraints in the MILP problem at each MILP call, of proofs generated by NeuralSAT and $\alpha\beta$-CROWN. We can see that NeuralSAT produces fewer subproofs, but with more complex MILP problems. In contrast, $\alpha\beta$-CROWN generates significantly more subproofs, but with simpler MILP problems. This variation suggests directions for future work, such as designing NeuralSAT to generate larger proof trees with simpler MILPs for better parallelization.

Fig. 4 explores the impact of MILP complexity and subproofs on APTPchecker performance. The MILP complexity distribution reveals distinct patterns across network architectures, e.g., FNN problems predominantly occupy the lower complexity range (0-1000 constraints), where APTPchecker nearly solves them all. In contrast, CNN problems have broader complexities, with many instances requiring more than 1000 constraints and exhibiting higher unsolved rates at greater complexities. For subproofs CNN and FNN architectures both generate remarkably similar proof tree sizes, with most problems requiring fewer than 1000 subproofs. However, the key difference lies in the complexity of individual MILP subproblems. FNN proofs involve simpler MILPs that can be efficiently verified, whereas CNN proofs remain challenging even with small proof trees due to the higher computational cost of solving individual MILP subproblems.

This architectural difference highlights that FNN problems generate more tractable MILP formulations, whereas CNN problems pose greater computational challenges due to their inherently more complex constraint structures.

### 5.3 RQ3: APTPchecker vs. Marabou's Checker

To evaluate proof checking performance in isolation, we compared APTPchecker against the proof checker built into the Marabou verifier [38], which uses Farkas's lemma and is implemented in the Imandra framework [25]. We used identical verification results from the Marabou verifier for both

Tab. 4: Proof checking times of Marabou's checker and `APTPchecker`.

| Checker | Proof checking time | | |
| --- | --- | --- | --- |
| | Mean | Median | Max |
| Marabou checker | 4 | 204 | 785 |
| `APTPchecker` | 3 | 9 | 38 |

checkers. `Marabou` successfully verified 54 problems from our benchmark suite. For each verified problem, we extracted the proof in `APTP` format and measured the time required by each checker to validate the proof.

Tab. 4 presents the proof checking times for both approaches. The similar mean times indicate that both checkers handle simple problems quickly. However, for more challenging proofs, `APTPchecker` demonstrates significant advantages with over $20\times$ speedup in both median (e.g., 9 vs 204 seconds) and maximum (e.g., 38 vs 785 seconds) checking times. This demonstrates the effectiveness of `APTPchecker`'s optimizations for complex proof validation.

### 5.4 RQ4: Unsound Cases

So far `APTPchecker` can validate sound proofs generated by existing verifiers. While unsound results are rare in practice, we were able to use this work to identify both real and synthetic examples of soundness bugs in existing verifiers, demonstrating its goal in ensuring verifier soundness.

**Real Bugs** One concrete example of such a soundness bug is documented in `NeuralSAT` Github issue #8, where `NeuralSAT` falsely returns UNSAT for a SAT instance [39].We were able to extract the `APTP` proof tree from this false verification result and analyzed it using `APTPchecker`. `APTPchecker` successfully found a valid counterexample, confirmed by `NeuralSAT` developers, that violates the claimed UNSAT result.

**Synthetic Bugs** Zhou et al. [40] explored verifier soundness by introducing synthetic bugs into verifiers. One studied class of bugs involves *randomly dropped branches* during BaB search, e.g., a proportion of new branches created in each BaB iteration are randomly discarded. This bug breaks the completeness of the BaB algorithm, as it may miss branches containing counterexamples, leading to false UNSAT claims. `APTPchecker` easily detects such incompleteness bugs because it always first checks is that all branches in the proof tree are properly visited and justified. Thus, `APTPchecker` will catch missing branches and raise errors indicating an invalid proof that it cannot certify.

## 6 Conclusion and Future Work

We introduce a proof format `APTP` which can express proofs generated by state-of-the-art DNN verifiers, and a proof checker `APTPchecker` that can validate proofs in this format. Together, these contributions establish a concrete foundation for *certifiable and reliable neural network verification*, closing the gap between practical verification and assurance required for real-world deployment.

**Limitations** `APTPchecker` relies on an external MILP solver such as Gurobi, which itself is not formally verified and can contain bugs (e.g., floating-point errors). While this is a limitation as we cannot eliminate all trusted components, this work allows us to significantly reduce the trusted code base. This can also be mitigated by using multiple solvers to cross-check results, e.g., CPLEX [41] and Xpress [42], or by replacing the MILP solver with a formally verified LP solver, e.g., SCIP [43].

**Potential Negative Societal Impact** The research line on DNN verification can exploited to find issues in DNNs and this work, which aims to improve DNN verification, indirectly supports that. However, DNN verification, and therefore this work, also helps to ensure that DNNs are safe and secure for deployment in critical applications.

## Acknowledgments and Disclosure of Funding

We thank the anonymous reviewers for their helpful comments. This work was supported in part by funds provided by the National Science Foundation awards 2129824, 2217071, 2501059, 2422036, 2319131, 2238133, and 2200621, and by an Amazon Research Award and an NVIDIA Academic Grant.

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

# A Syntax and Grammar of APTP

Fig. 2a in §3.2 outlines the APTP syntax and grammar, represented as production rules. A proof is composed of *declarations* and *assertions*. Declarations define the variables and their types within the proof. Specifically, *input variables* (prefixed with X) and *output variables* (prefixed with Y) are declared as real numbers, representing the inputs and outputs of the network. Additionally, *hidden variables* are declared with specific piece-wise linear (PWL) activation functions, such as ReLU. These hidden variables correspond to the internal nodes of the neural network that process the input data through various activation functions.

Assertions are logical statements that specify the conditions or properties that must hold within the proof. Assertions over input variables are *preconditions* and those over output variables are *post-conditions*. Each assertion is composed of a *formula*, which can involve terms and logical operators. Formulas include simple comparisons between terms (e.g., less than, greater than) or more complex logical combinations using and and or operators. The terms used in these formulas can be variables or constants.

The declare-* statements declare input, output, and hidden variables, while the assert statements specify the constraints on these variables (i.e., the pre and postcondition of the desired property). The hidden constraints represent the activation patterns of the hidden neurons in the network (i.e., the proof tree). Each and statement represents a tree path that represents an activation pattern.

# B Correctness of APTPchecker Implementation

We were able to verify the implementation of the core APTPchecker algorithm (§4.1) using the CrossHair [36] symbolic execution for upto certain thresholds (e.g., timeout per condition per_condition_timeout=10).

To perform such analysis, we need to create a simplified version of APTPchecker including: (1) No optimization – remove all optimizations in Apdx. C; (2) Assume that Gurobi (MIP) is correct, therefore, the condition indicating whether MIP is correct or not must be made up (e.g., $sum(n) \geq 0$ – summation of all literals (e.g., variable and branch condition) in a leaf node); and (3) Add pre- and post-conditions to the main function. This make APTPchecker codebase minimal with just about 100 LoC. In particular, some pre- and post- conditions are listed in Listing 1.

```
1 """
2 pre: isinstance(proof, list)
3 pre: all(isinstance(p, list) for p in proof)
4 post: _ in {CERTIFIED, UNCERTIFIED}
5 post: (_ == UNCERTIFIED) == (any(sum(n) < 0 for n in proof))
6 post: (_ == CERTIFIED) == (all(sum(n) >= 0 for n in proof))
7 """
```

Listing 1: Pre- and Post- Conditions for CrossHair

CrossHair outputs are shown in Listing 2.

```
1 attempt_call() Postcondition confirmed.
2 analyze_calltree() Path tree stats {CONFIRMED:58}
3 analyze_calltree() Iter complete. Worst status found so far: UNKNOWN
4 analyze_calltree() Exceeded condition timeout, stopping
5 analyze_calltree() Aborted calltree search with UNKNOWN and 0 messages. Number of
      iterations:  58
6 analyze_class() Analyzing class  ProofReturnStatus
7 condition_parser() Using parsers:  (AnalysisKind.PEP316, AnalysisKind.icontract,
      AnalysisKind.deal)
8 analyze_class() Analyzing class  ProofTree
9 condition_parser() Using parsers:  (AnalysisKind.PEP316, AnalysisKind.icontract,
      AnalysisKind.deal)
10 analyze_function() Analyzing  mip_worker
11 condition_parser() Using parsers:  (AnalysisKind.PEP316, AnalysisKind.icontract,
      AnalysisKind.deal)
```

Listing 2: CrossHair traces

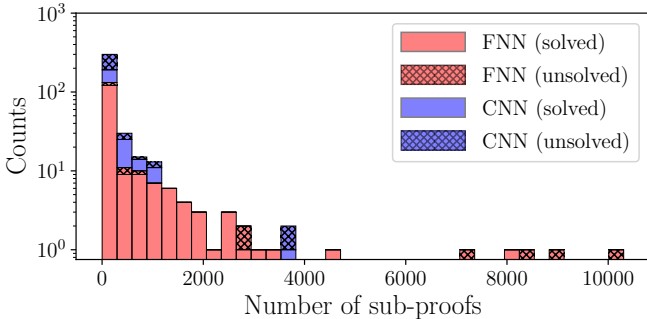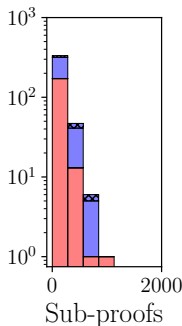

Fig. 5: Number of sub-proofs per problem without (left) and with (right) `APTPchecker` optimizations.

## C   Optimizations

While the core `APTPchecker` algorithm in Alg. 2 is minimal, it can be inefficient. `APTPchecker` employs several optimizations to improve its efficiency. These are crucial for checking large proof trees generated for challenging problems.

**Neuron Stabilization**   A primary challenge in DNN analysis is the presence of large numbers of piece-wise linear constraints (e.g., ReLU) which generate a large number of branches and yield large proof trees. In the MILP formulation, this creates many disjunctions which are hard to solve. To reduce the number of disjunctions, `APTPchecker` uses *neuron stabilization* [9] to determine neurons that are *stable*, either active or inactive, for all inputs defined by the property pre-condition. For all stable neurons, the disjunctive ReLU constraint is replaced with a linear constraint that represents the neuron's value. This simplifies the MILP problem.

`APTPchecker` traverses the DNN and computes stable neurons. It initializes the MILP model with input constraints and then iterates over each layer of the network. Next, for each layer, it creates constraints depending on the layer type. Moreover, it uses approximation to estimate bounds of neuron values to determine neuron stability. Next, it filters unstable neurons and attempts to make them stable by optimizing either their lower or upper bounds.

**Pruning Leaf Nodes**   `APTPchecker` uses a backtracking mechanism to check the parent node only when the child nodes are infeasible. Specifically, if it determines unsatisfiability of leaf $l$, it will check the parent $p$ of $l$. If $p$ is unsatisfiable it immediately removes the children of $p$ (more specifically the sibling of $l$). Next it backtracks to the parent of $p$ and repeats until meeting a stopping criteria. This optimization reduces the number of LP problems that need to be solved, making the proof checking process more efficient.

**Parallelization**   `APTPchecker` leverages the structure of APTP proof tree to parallelize the checking of leaf nodes. Each tree path is an independent sub-proof and partitions of the tree allow checker to leverage multiprocessing to check large proof trees efficiently.

### C.1   Proof Checking Optimizations

Fig. 5 (left) plots a histogram of the number of sub-proofs solved per verification problem, i.e., the number of nodes of the proof tree. When interpreting these plots, understand that the y-axis log scale means that vertical distances have a different meaning as you move upward in the plot. While the vast majority of the verification problems have proof trees of fewer then 2000 leaves, but 17 of them have larger trees up to a maximum of more than 10000 leaves. Note also that even among the smaller sized proof trees, there are some problems that cannot be solved. This is due to complexity of solving the MILP constraints at the leaves of those proof trees.

Fig. 6 (left) plots a histogram of the number of occurrences of MILP problems of a given complexity across the benchmarks. Here again we see a spread in data, but unlike with the number of sub-proofs the CNN benchmarks seem to have consistently larger constraints and there is a clear bias among the

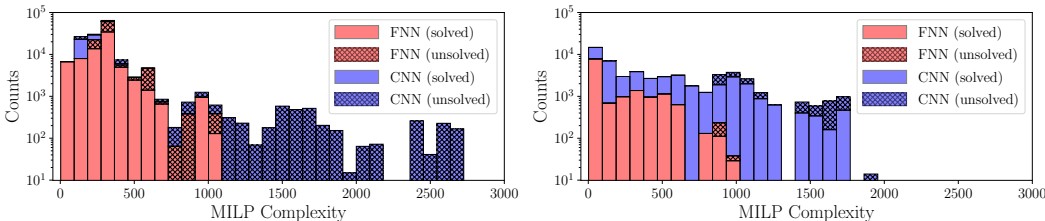

Fig. 6: Number of constraints per problem without (left) and with (right) `APTPchecker` optimizations.

unsolved problems towards larger constraint size. To optimize proof checking, we must address both of these sources of complexity.

The Fig. 5 (right) and Fig. 6 (right) explore the impact of the S and X optimizations on the number of sub-proofs and MILP complexity. Across the benchmarks optimizations reduce the number of sub-proofs is to less than 1000 and MILP complexity to less than 2000. The reduction in sub-proofs directly contributes to the increase in performance of `APTPchecker`, but the reduction in MILP complexity is more subtle. Integer programming, and thus MILP, is known to be NP-Hard in general [44]. The stabilization optimization addresses this complexity by calculating sets of variables that are forced to take on specific values based on other constraints in the MILP problem. For each such variable, the constraints associated with it is effectively eliminated. We can observe this in comparing the left and right of Fig. 6 where we see both constraints of higher complexity eliminated and the peak of the constraint distribution shifted downward from 400 to 100 constraints.

## D  Related Work

Proof checking has been widely-recognized in the field of constraint solving such as SAT/SMT solving. (e.g., [45, 46, 47]). There is extensive literature on clausal proof generation and checking for SAT solvers [48, 49, 50, 10, 51]. Most modern SAT solvers can produce resolution-based proofs in standard formats (e.g., DRAT [10]), which can be independently checked by proof checkers, e.g., by efficient, untrusted programs such as `DRAT-trim` [10] or by certified, slower programs that work on extended formats such as LRAT [19] and GRAT [12].

SMT proof checkers [52, 53, 54] share the same purpose of checking unsatisfiability proofs, but they are more complex than SAT proof checkers due to the richer languages and theories of SMT formulas (e.g., theory of strings). Two significant proof-producing state-of-the-art SMT solvers are z3 [20] and veriT [21] that both can have their proofs successfully reconstructed in proof assistants [55, 56, 57, 58]. Other proof-producing SMT solvers are MathSAT5 [22] and SMTInterpol [23], CVC5 [24] and CertiStr [59]. Recently, a high-performance stand-alone checker Carcara [60] for the Alethe [61] proof format was also introduced.

Compared to SAT/SMT, DNN verification is a relatively new field, and the development of proof checkers for DNN verifiers is few. To the best of our knowledge, there is only one line of work [25] that is explicitly for the `Marabou`. This work uses Farkas's lemma [62] for checking and is implemented in the Imandra [26] that can produce verifiable code. Our work generalizes to neuron-splitting based DNN verification and introduces a new, more expressive proof format, APTP, that can be adopted by other DNN verifiers. Our proof checker, `APTPchecker`, is also significantly more capable (§5.1).

