# OpenReview forum: "Generating and Checking DNN Verification Proofs"
_NeurIPS.cc/2025/Conference — NeurIPS 2025 poster_

### Official Review · Reviewer_r99i · 2025-07-01

**Clarity:** 4
**Significance:** 2
**Originality:** 2
**Rating:** 5
**Confidence:** 3

**Summary:**

This paper presents APTP, a language for expressing proofs of correctness of
neural networks, and APTPchecker, a tool for verifying proofs written in APTP.
This allows neural network verification tools to grow in complexity while still
producing a proof object which is checkable by a small trusted verifier,
increasing the trustworthiness of deep learning. APTP is specialized to proofs
based on branch-and-bound neuron-splitting techniques and mixed integer linear
programming. This proof structure is common to many state-of-the-art neural
network verification tools. APTP encodes the branches generated by these
verification tools and then APTPchecker reconstructs the MILP instances needed
to verify the property in question along each branch. In experiments, existing
verification tools are modified to produce APTP proofs and it is shown that
APTPchecker is able to verify these proofs more effectively than existing proof
verifiers for neural networks.

**Questions:**

- How well does APTPchecker work with Marabou as a verification engine? It would
  be interesting to see how APTPchecker compares to Marabou checker when using
  the same underlying verifier.

**Ethical Concerns:**

["NO or VERY MINOR ethics concerns only"]

**Final Justification:**

APTPChecker presents a valuable contribution to the literature by providing a small, trusted core that can be used to verify proofs generated by more complex tools. There is a small weakness in that APTPChecker cannot handle activation functions that are not piecewise-linear (sigmoid, tanh, etc.), but I believe this is not a major problem given how commonly networks use ReLU activations. The additional experiments presented during the discussion show that APTPChecker is efficient even compared to a checker specifically designed to work with Marabou.

**Limitations:**

yes

**Quality:**

3

**Strengths And Weaknesses:**

Strengths:

- The utility of a small, trusted proof checker has been established in
  classical verification and bringing this idea to DNN proofs is a useful
  contribution.
- The experiments show that APTPchecker is able to check proofs generated by
  existing state-of-the-art DNN verification tools.
- The APTP proof language is generic enough to be used by several different
  solvers.

Weaknesses:

- The proof language is limited to proofs based on branching on ReLU activation
  status. To my understanding, this aligns with many existing verification
  tools, but it precludes the use of APTP with non-ReLU activation functions or
  verifiers based on other algorithms.
- Some notation in the results section could be explained. For example, I don't
  think the S+X label used in the graphs is explained anywhere, though I gather
  it refers to the optimizations that APTPchecker can use.

---

> ### Author Rebuttal · Authors · 2025-07-30
>
> Thank you for your encouraging comments. We provide our responses to your questions and concerns below.
>
> 1. Compare to Marabou checker
>
>
>
> We were able to extract APTP proof trees from all 54 problems that Marabou could verify and check proofs.
> - Marabou checker (min/median/max): 4/204/785 (seconds)
> - APTP checker (min/median/max): 3/9/38 (seconds)
>
> Note that compared to NeuralSAT/α-β-CROWN, Marabou fails to verify many problems. For those that it can solve, Marabou checker is a lot slower than APTP checker as shown.
>
> 2. APTP format limited to ReLU
>
> We will clarify in the paper that the APTP format is not restricted to ReLU and can handle any piece-wise linear activations as hidden status can be expressed by logical formulae. For example, for Leaky_a(x) = { x if x ≥ 0 else ax }, it status are expressible by APTP, e.g., (>= Ni 0) for on and (< Ni 0) for off, which is essentially identical to ReLU, while MIP encoding will handle the Leaky operation.

---

> > ### Comment · Reviewer_r99i · 2025-08-04
> >
> > Thank you for the response. I remain convinced that APTPChecker is a valuable contribution, and I'm happy to stand by my original score.

---

> > > ### Author Response · Authors · 2025-08-07
> > >
> > > Thanks. We appreciate the review and feedback, and will incorporate your comments in the paper revision

---

### Official Review · Reviewer_iqSc · 2025-07-02

**Clarity:** 4
**Significance:** 3
**Originality:** 3
**Rating:** 5
**Confidence:** 5

**Summary:**

In NN verification, verifiers return "safe" or "unsat" for properties they have verified. However, due to the complexity of these tools, there's an inherent risk that they contain bugs and may be unsound. To verify the verifier, the authors propose to extract a proof-tree from the verifier. This represents the leaf nodes of the branch-and-bound tree, defined by the chosen branching decisions (constraints on internal neuron activations). If an external verifier can confirm that all those leaf nodes define safe subproblems, and their union is the original query, the correctness of the proof has been shown.
The authors propose APTPchecker as such a tool: By implementing the logic in Gurobi, and partially verifying the implementation using symbolic execution (CrossHair), they minimize the risk that APTPchecker is unsound.

In the experimental section, the authors demonstrate that APTPchecker can verify the correctness of a large portion of the proofs generated by two different verification tools, with abCROWN being the winner of several VNN-COMP iterations. This demonstrates the real-world applicability of their proposal.

**Questions:**

Is it possible to extract a proof tree from Marabou, that you could verify using APTPchecker? If so, how does APTPchecker compare to the built in proof verifier in Marabou?

**Ethical Concerns:**

["NO or VERY MINOR ethics concerns only"]

**Final Justification:**

The authors have responded to all questions and were able to provide additional strong experimental results.

**Limitations:**

Yes

**Quality:**

3

**Strengths And Weaknesses:**

*Strengths*
The paper discusses an important aspect of NN verification: How can we trust the verifiers? The paper does a very good job of introducing the relevant concepts and motivating the need for their proposed algorithm.
There is very limited prior work in this regime, so it is great to see some progress on this.
The experiments demonstrate the practical applicability and show that APTPchecker can in fact verify the proofs generated by two different NN verifiers.
The description of the optimizations (in the appendix) will be helpful for other researchers to replicate the results.

*Weaknesses*
I would have liked to see an evaluation on more benchmarks/network architectures. How does APTPchecker perform for more complex verification queries? E.g. for the benchmarks from one of the previous VNN-COMP iterations, which of them are already supported, and how toes APTPchecker perform there?


Minor
- Figure 6: I assume the dashed lines are pure abCrown and NS, without the APTPchecker?
- Should the sentences regarding CrossHair be moved to the Correctness section?
- Line 48 "emit[s]"
- Grammar: Line 321-322

---

> ### Author Rebuttal · Authors · 2025-07-30
>
> Thank you for your encouraging feedback. We provide our responses to your questions and concerns below.
>
> 1. More complex networks
>
> APTPchecker supports FNNs and CNNs but not ResNet.  We mentioned in the paper (Sect.5) that this is an engineering limitation rather than a fundamental algorithmic one.  For ResNet, the skip connections would require extending our MILP encoding, but the proof checking logic (validating that claimed infeasible branches are indeed infeasible) remains unchanged. The core APTP proof checking approach extends naturally to more complex architectures.
>
> In the last several days we were able to do just that and support Resnet. However, we did not have enough time to run an experiment with it on Resnet benchmarks (we spent most time on extracting Marabou’s code as requested below by you and another reviewer).  We'd be happy to add additional results during discussion time if allowed.
>
> 2. Compare to Marabou checker
>
>
> We were able to extract APTP proof trees from all 54 problems that Marabou could verify and check proofs.
> - Marabou checker (min/median/max): 4/204/785 (seconds)
> - APTP checker (min/median/max): 3/9/38 (seconds)
>
> Note that compared to NeuralSAT/α-β-CROWN, Marabou fails to verify many problems. For those that it can solve, Marabou checker is a lot slower than APTP checker as shown.

---

> > ### Comment · Reviewer_iqSc · 2025-08-04
> >
> > Thank you for your response and the work you put into running the additional experiments!
> >
> > The new results support your approach, and would be a great addition to the paper. I stand by my original rating.

---

> ### Author Response · Authors · 2025-08-04
>
> thanks!  yes, we will include these results to the paper.

---

### Official Review · Reviewer_Pxf9 · 2025-07-03

**Clarity:** 3
**Significance:** 4
**Originality:** 3
**Rating:** 5
**Confidence:** 5

**Summary:**

This paper introduces Activation Pattern Tree Proofs (APTP), a verifier-independent proof format for neural network verification, along with APTPchecker, an efficient and formally correct proof checker. By enabling scalable and reliable validation of verification results, the approach aims to increase trust in neural network verification tools, which are essential for deploying DNNs in safety-critical systems.

**Questions:**

- I would be interested in seeing a table comparing the runtime required to verify with alpha-beta-CROWN and the runtime your toolbox requires to certify the proof. Could you include such a comparison?

- It would also be helpful to see a table reporting the size and complexity of the DNNs evaluated in your experiments. Please consider including this information.

**Ethical Concerns:**

["NO or VERY MINOR ethics concerns only"]

**Final Justification:**

I have revised my review based on the discussion in support of acceptance and under the assumptions that (1) the authors add an example that demonstrates the importance of proof checking in neural network verification, eg, with a demonstration on an example from VNN-COMP where tools disagree on results, or a floating-point precision example from the literature, which will more clearly demonstrate the importance of this work, (2) clarify the contributions in the paper as discussed especially for the machine learning audience, and (3) make the other improvements that have arisen in the discussion. The potential impact of this direction is significant, with a reasonable approach presented in the paper for proof checking in neural network verification, which is clearer now from the discussion and that addressing these points in a revision if accepted will improve the paper and its potential impact, and that may especially be appreciated by the machine learning audience.

**Limitations:**

Yes

**Quality:**

3

**Strengths And Weaknesses:**

Strengths:

- The paper is well written and clearly describes the proposed approach.
- The proposed approach addresses proof generation and checking in the context of neural network verification, which has not significantly been addressed in the literature. This is an important consideration in neural network verification, where soundness errors have been demonstrated somewhat (e.g., tools may say a specification holds when it does not). As such, the paper presents contributions to eliminating this issue to ensure verification results hold (e.g., so if a tool says a specification holds, it actually does).
- The results illustrate the approach with a sufficiently rigorous empirical evaluation, albeit it has some limitations.

Weaknesses:

- The contribution of the work is not clearly articulated, particularly for a machine learning audience. While some aspects of how tools could produce incorrect results is discussed, it could be much more strongly motiviated. For example, with concrete examples. There are other examples in the literature that exist (eg, due to floating point issues and related numerical issues).

- Some aspects of the intuition of the proof certificate and tree (eg Figure 1) do not fully convey the point of the approach, especially for a machine learning audience. While a verification reader can appreciate this, a machine learning one may not, and this example in particular could be better motivated, as it may come across simply as existing known BaB approaches.

- The proposed method essentially re-derives results using MILP and LP techniques that are very similar to those already employed by existing approaches. Could you clarify what your work contributes technically to the verification problem? The LP and MILP formulations you describe are well-known techniques commonly used in established branch-and-bound methods such as alpha-beta-CROWN.

- The number of constraints required to certify the proof appears to grow exponentially with the size of the DNN. Could you explain how you address scalability in this context?

- The procedure you use to certify proofs seems very similar to the verification steps already performed by existing tools. It appears you are repeating these steps without introducing substantial novelty. Could you please elaborate on what you see as the core contribution of your approach?

- The presentation of Figures 3 and 6 can be improved: the legend overlaps much of the core content of the figures, making them difficult to read.

---

> ### Author Rebuttal · Authors · 2025-07-30
>
> Thank you for your time and reviews. We provide our responses to your questions and concerns below.
>
> 1. Exponential Constraint Growth.
>
> Yes! The number of constraints can indeed grow exponentially, but APTP already includes mechanisms to reduce them with neural stabilization, tree pruning, and parallelization. Our experiments show that the numbers of proofs (and constraints generated) are moderate, e.g., 601 sub-proofs to be checked on average for proofs generated by NeuralSAT.
>
> 2. Contributions of Proof checking over verification
>
> APTP makes the following contributions:
> - APTP validates that verifier-claimed infeasible branches are indeed infeasible, rather than searching for satisfying assignments as the verifier does.
> - Verifiers like α-β-CROWN and NeuralSAT have many additional steps, optimizations, and heuristics beyond the minimal BnB algorithm presented. In contrast, APTP strictly follows BnB and is independent of all those additional complexities: it only needs to validate the final activation pattern claims, regardless of the sophisticated search strategies, bound tightening techniques, or pruning heuristics used by the verifier.
> - Our key contribution is separating proof generation (done by existing verifiers) from proof checking (done by our minimal checker). This provides:
>     - Reduced Trusted Computing Base: 800 SLOC checker vs 20K+ SLOC verifiers.
>     - Verifier Independence: APTP format works with multiple verifiers (α-β-CROWN, NeuralSAT).
>     - Independent Validation step of verification results
>     - Checker validation: We attempt to prove the checker itself correct through symbolic execution (Section 6.3 and Appendix B).
>     - Multiple solver support: Because we use the MILP solver as a black box, multiple solvers can be employed to increase confidence in results.
>     - The APTP format captures activation pattern trees in a verifier-independent way that enables efficient checking while remaining compact and human-readable.
>     - Our optimizations target proof checking efficiency (stabilization, pruning, parallelization) rather than verification search efficiency, leading to different algorithmic choices.
>
> 3. Runtime table for α-β-CROWN.
>
> We'd be happy to. We have all the data to provide a complete table but they took too much space to include in the paper.  We will include all data in the appendix. Here is a summary of the runtimes
>
>
> - FNN (min/median/max):
>     - Verify (4.0/9.0/481.1)
>     - Proof check (3.4/19.0/869.7)
> - CNN (min/median/max):
>     - Verify (6.4/16.4/50.1)
>     - Proof check (12.7/67.4/866.1)
>
> In short, the results are similar for α-β-CROWN as expected: verification time is a lot shorter than checking verification results.   While proof checking is more expensive than verification, we emphasize that generally speaking one would only check the proofs once.   During development as properties and models are evolving one will rerun verification multiple times, but at the very last step when all the properties have proofs that is the time that you will check them.   In this context the increased cost of proof checking is not as much of a concern.
>
> Note that in our paper we show these results for NeuralSAT in figure 3, but figures are explicitly not allowed by NeurIPS rebuttal.
>
>
> 4. It would also be helpful to see a table reporting the size and complexity of the DNNs evaluated in your experiments.
>
> We believe Tab.1 (page 7) provides these details of the DNNs. Let us know if you need additional info.

---

> > ### Comment · Reviewer_Pxf9 · 2025-08-04
> >
> > Thank you for your responses. I agree this direction is important, and the response illustrates some clarity of the contribution, as well as addressing clarifications. These questions in part were stated to motivate the importance of presentation for the audience, that I hope will be carefully taken into account in the presentation, if accepted.
> >
> > As also arose though in some other reviews, the point of the approach could still be better demonstrated, e.g., with an example that demonstrates the benefit of checking proofs. If all the existing methods are demonstrated to be sound, that is great (and important from verification viewpoint), but still from a machine learning viewpoint, so what?
> >
> > Adding some example. There are some unsound results that have been reported in the literature (eg, floating point) and some examples at VNN-COMP where tools have disagreed on results and some tool is unsound. If accepted, I think it is essential to add some example that illustrates such soundness problems, eg, that the approach can detect these, which would much more strongly demonstrate the point of the paper.
> >
> > Very concretely, in VNN-COMP, there are already mechanism to validate counterexamples, eg, check if a counterexample is a real counterexample or not (up-to some numerical tolerance). However, there can be cases where the final result truly is unknown, due to lack of ground truth (eg, say one tool says unsat and another unknown, it could be the case the unsat tool just guessed unsat, and it really is unknown), or, due to numerical precision, something could actually be wrong (eg, it could actually be sat). Some example with mismatched tool outputs should be taken and validated/invalidated with the approach, given the claims of tool agnostic support and to further demonstrate this claim.
> >
> > In summary, I am in favor of acceptance, assuming the authors address this point if accepted to really get across the point of doing the proof checking, especially for this audience (and also eg that would be great to incorporate into VNN-COMP or related initiatives, as has been done at SAT-COMP and others, for instance, one of the distinguished papers at CAV'25 was on this topic of incorporating proof checking into SAT-COMP that yielded both performance benefits and these guarantees, which illustrates the potential impact this type of work can have in this domain).

---

> > > ### Author Response · Authors · 2025-08-05
> > >
> > > Thank you for your comments. We will emphasize the motivation in the introduction and we will include a section with concrete examples of unsoundness to demonstrate the work.  For this section, we will do the following:
> > >
> > > First, we'd like clarify that DNN bugs fall under 2 categories: (i) unsound, in which the DNN violates the property (ground truth SAT, a CEX exists), but the verify claims it does not (UNSAT, no CEX) and (ii) false positive, where the DNN has the property (ground truth UNSAT), but the verifier claims CEX exists (SAT).
> > >
> > > The 2nd category of false CEX, as you have pointed out, can already be handled by checking the reported CEX (e.g., VNN-COMP already does this, and indeed has caught many such bugs -- usually due to numerical and floating point rounding errors).
> > >
> > > Our tool specifically targets the 1st category--unsoundness.  VNN-COMP lacks mechanism to check for UNSAT claim by verifiers, and hence the motivation for our work to encourage verifiers to produce UNSAT proofs and provide an independent checker for those.
> > >
> > >
> > > Second, we agree a concrete example of our tool detecting unsoundness bug would be helpful, and we have found a real unsound bug in the NeuralSAT tool that we will use to demonstrate the effectiveness of our work. In particular, its Github issue #8 (we cannot directly post a link due to NeurIPS' rules) shows---and NeuralSAT developers confirmed---that NeuralSAT returns UNSAT in an instance with SAT ground truth. We looked a bit deeper and this bug is likely due to some optimizations or heuristics in causing incorrect over-approximation in NeuralSAT and thus causing the tool to return UNSAT for a SAT instance. This seems to be a non-trivial bug as NeuralSAT has been through many VNN-COMP benchmarks and yet still miss it. APTP will be able to detect this UNSAT result (proof) that NeuralSAT produces because it does not rely on any specific optimizations or heuristics of verifiers and mainly encodes result proof trees as MILP constraints and solves them. Thus in this case APTP will find a CEX to disprove this UNSAT result (in other words, detect error in the UNSAT proof).
> > > We will use this an example and provide details to showcase the practicality of our work.
> > >
> > > Finally, we completely agree and will work on getting this incorporated to VNN-COMP as it encourages verifiers to include proofs when they claim soundness (currently they do not and our works show they can and it's not hard to do so) and provide an additional layer of independent checking to reported results.

---

> > > > ### Comment · Reviewer_Pxf9 · 2025-08-06
> > > >
> > > > Sounds great, thanks for your response and great work!

---

### Official Review · Reviewer_my7d · 2025-07-03

**Clarity:** 3
**Significance:** 2
**Originality:** 2
**Rating:** 3
**Confidence:** 4

**Summary:**

The paper proposes the generation of verification proofs by neural networks (which capture the sequence of split decisions and branch evaluations made), and puts forward a MILP-based procedure for validating these proofs. The empirical evaluation of this focuses on performance comparisons with previous work and applicability demonstrations with respect to different neural network verifiers.

**Questions:**

Please discuss the three points above pertaining to the evaluation of your method.

**Ethical Concerns:**

["NO or VERY MINOR ethics concerns only"]

**Final Justification:**

The paper provides limited evaluation in that it does not make clear the range and scale of models that the proposed approach can support.

**Limitations:**

The second point described in "Strengths and Weaknesses", which is potentially a limitation of the approach, has not been addressed. The potential negative societal impact has been adequately addressed.

**Paper Formatting Concerns:**

I have not noticed any major formatting concerns.

**Quality:**

2

**Strengths And Weaknesses:**

While the paper makes an important observation concerning the practical unsoundness of theoretically sound verifiers, thereby adequately motivating the need to ensure the correctness of certification results, the framework proposed appears preliminary for NeurIPS. This is particularly the case for evaluation of the method which in my opinion has two main shortcomings.

First, it considers only toy models and does not show the scalability of what proposed to the bigger models that neural network verifiers can handle. This is important as models and specifications that require a lot of branching (with branches often efficiently analysed on GPU) may prove problematic for the slower CPU-based MILP solvers. Further, it is not clear that the overall runtime (of verifying and the validating verification proofs) is better than simply verifying the underlying models using efficient MILP-based solvers, such as [1, 2].

Second, the paper does not make clear that certified verification proofs ensure the correctness of the verification result. In particular, the paper does not only assume that the underlying MILP solver is correct, but also that the MILP-compilation the paper provides is correct. Efficient compilations however crucially depend on tight bounds for the ReLU nodes, which are typically computed by bound propagation schemes, a core component of  neural network verifiers that is susceptible to floating-point errors. Further, the paper discusses only ReLU splitting and does not address input splitting, a branching strategy that is also present in most neural network verifiers.

Third, the paper does not provide any concrete examples of unsound results from neural network verifiers which have been proven unsound by the proposed method.

[1] V. Tjeng, K. Xiao, R. Tedrake. Evaluating robustness of neural networks with mixed integer programming. ICLR19.

[2] P. Kouvaros, A. Lomuscio. Towards Scalable Complete Verification of ReLU Neural Networks via Dependency-based Branching. IJCAI21.

---

> ### Author Rebuttal · Authors · 2025-07-30
>
> Thank you for your critical review. We’ve carefully considered your comments and provide detailed responses below to address raised concerns.
>
>
> 1. "Toy" models and not scale to networks that current verifiers can handle.  Large models require lots of branchings.
>
>
> We respectfully disagree that our evaluation benchmarks are “toy models”.  Our benchmark consists of 400 problems across DNNs with up to 1.7M parameters, which are certainly not toys, even for SoTA DNN verifiers such as ABCrown (e.g., see VNN-COMP'24 results on how these tools struggle on similar or even smaller sizes).  Moreover, the only existing proof checking work (Marabou's) evaluates on significantly smaller ACAS Xu networks with just 13K parameters.
>
> The number of constraints created due to large branching indeed is theoretically exponential. However, in practice we do not observe that and more importantly APTP includes various techniques to improve scalability (e.g., neuron stabilization, proof tree pruning, and parallelization) that are shown to be effective in the paper (e.g., 601 proofs in average for proofs generated by NeuralSAT).
>
>
>
> 2. Overall runtime (of verifying and the validating proofs) is better than simply verifying the underlying models.
>
>
> We are confused by this question.  The purpose of this work is to provide support for verifying the verifier, i.e., we check that reasoning steps of existing DNN verifiers are sound.  Thus our work does not aim to compete with verifiers but rather is used *after* verification to ensure the correctness of the steps performed by the verification implementation.
>
> 3. Correctness of MILP solver
>
> We discussed this in the paper (4.1.2): APTP uses the Gurobi MILP solver as a blackbox, and as with all other verification work, we do not prove its correctness.  However, our main goal with APTP is *not* to eliminate all trusted components, but rather to significantly reduce the size of the trusted code base (TCB) necessary to assure that the verification run was sound, so that users can have increased confidence in the result. APTP makes the following contributions toward improving confidence in ML verification:  (i) it sets up an architecture to allow for proof checking that is separate from the complex algorithms for DNN verification, allowing their results to be checked independently;   (ii) it implements an independent proof checker APTPChecker and goes further by verifying the implementation of the checker itself—this level of formalism and assurance is rarely seen in DNN verification work and reduces the TCB; (iii) the APTP architecture allows for targeting further reductions in the TCB, e.g., in the future the developers can plug in a different MILP solver, to provide increased confidence under the assumption that MILP solver implementations will not be strongly correlated, or replace LP solving with a trusted theorem prover.
>
> We also note that the Marabou checker also relies on the correctness of the blackbox SMT solver used inside Imandra (and if we want to go further it relies on the correctness of its integration of Ocaml). This reliance on solvers (e.g., MILP solver in our case) is a common and accepted trade-off in the SoTA of verification checking.
>
>
> In summary, while APTP does not eliminate *all* trust assumptions, it significantly reduces them compared to prior work, providing a framework for DNN verification proof checking.  We believe it lays a path toward more trustworthy DNN verification that the community can build on and provides a level of verification of verification results that does not currently exist in the literature.
>
>
> 4. Floating-point (fp) errors due to abstraction/bound computations.
>
> It's a good observation! Indeed, APTP can have fp errors due to the initial bound computation using interval abstraction. However, we do not rely on bound propagation but instead on neuron stability solved by MILP, which is more robust to fp errors compared to complex bound propagation schemes.
> We note that the initial interval abstraction can be completely replaced by the MILP formulation (Equation 2), e.g., encoding ReLU and solving bounds directly (in fact, the current implementation of APTP does just that and therefore resolves this concern).  We also note that the Marabou checker work also relies on abstraction (DeepPoly) and has the same fp concern.
>
> Interestingly, from what you said, verification tools could produce unsound results due to the mentioned fp error issue (due to abstraction and optimizations).  If this happens then APTP checker can catch it because it does not have such optimizations and abstractions and instead just recheck their work using MILP solver.
>
> 5. Input splitting
>
>
> We did not consider input splitting which helps deal with networks with low dimensional input spaces. We focus on the major BaB architecture for neuron splittings as challenging and interesting problems since those techniques are used by SoTA NN verifiers and are triggered when verifying networks with lots of neurons and high dimensional inputs.
>
> That said, APTP proof format can be extended easily to handle the input splitting cases where each formula encoding hidden activation is converted to input intervals capturing the input subspace being checked. The core of APTPchecker would remain unchanged as would the level of trust one could place in verification results.
>
> 6. Concrete examples of unsound results from neural network verifiers which have been proven unsound by the proposed method.
>
> While we acknowledge this would strengthen the empirical motivation, it does not diminish our contribution's significance. Our evaluation demonstrates that the two state-of-the-art verifiers we tested (α-β-CROWN and NeuralSAT) are sound for the problems we examined - APTPchecker successfully validated 363 out of 400 verification results with no cases where a verifier incorrectly claimed UNSAT.
>
>
>
> The primary goal of APTPchecker is to provide independent validation that increases confidence in verification results. When APTPchecker cannot validate a proof, this would lead developers to further investigate whether unsoundness occurred.

---

> > ### Comment · Reviewer_iqSc · 2025-08-04
> >
> > I am not the original reviewer, but have a comment regarding
> > > Overall runtime (of verifying and the validating proofs) is better than simply verifying the underlying models.
> >
> > The way I understand this question is as follows: It might be that verifying the instance with e.g. NS, and then verifying it with your tool takes longer than to just encode the entire problem instance as one MILP problem and verifying that. The latter would be sound by definition (because you assume that MILPs are solved correctly by Gurobi), so your tool would have no benefit.
> >
> > Note that I do not expect this to be true. Your results already demonstrate that your overhead is limited. On the other hand, if Gurobi were sufficient to verify the instance by itself, this would be used more often.
> >
> > However, including such an experiment should be easy to set up, and may alleviate this criticism.

---

> > > ### Author Response · Authors · 2025-08-04
> > >
> > > Thanks for the clarification.  We have done a quick experiment as suggested, i.e., encoding the DNNV problem as a native MILP formulation and solving it directly using Gurobi. Indeed, Gurobi cannot solve the MILP formulation of large problems (e.g., CNN-based, timeout after 1000s), but it works fine for smaller networks (e.g., FNN-based).
> > >
> > > However, regardless of our results above, we still want to clarify that the goal of our approach is to check results of *other* verifiers. We do not aim to create an efficient DNN verifier (Crown/NeuralSAT, etc already exist), but to provide an independent checker for verifiers. In other words, we require the results/proofs of the verifier (e.g., the  branches/constraints they considered) and check that their reasoning steps are correct.  Our inspiration stems from SAT competitions where competing tools have to provide `unsat` proofs, which are then verified by independent checkers (e.g., drat).

---

> > ### Comment · Reviewer_my7d · 2025-08-05
> > **Thank you for the response**
> >
> > I sincerely thank the authors for the comprehensive response on which I have some follow-up questions. I want to clarify that I will be willing to update my score if these are addressed.
> >
> > **Concern on models and comparisons with MILP-based solvers.**
> >
> > - While I agree with the authors that in the context of neural network verification the models are not "toy" models, my concern remains. Unless I am missing something, the evaluation considers three benchmarks: CIFAR2020, MNISTFC and MNIST_GDVB. As far as I know CIFAR2020 and MNISTFC are some of the smaller benchmarks to participate in VNNCOMP. For instance MNISTFC has three models, having 2, 4 and 6 layers, each with 256 ReLUs. In earlier versions of VNNCOMP (VNNCOMP 21) the verifiers can effectively verify both MNISTFC and CIFAR2020 - since then, most verifiers have made significant advances, thus their performance on the benchmarks is expected to be even better. I am not aware of MNISTFC_GDVB, the paper does not report the specific architectures and perturbation radii used, and I could not find that information in the citation given - perhaps I have missed it? It is thus currently my view that the evaluation did not consider a sufficiently wide range of benchmarks, covering different depths, layer sizes and ReLU counts.
> >
> > - I think that the verification difficulty mostly depends on the number of non-linear components present in the network, rather than the number of tunable parameters - for instance, a network comprising of only big affine layers should be easy to verify with an LP solver - even if the network has lots of tunable parameters.
> >
> > - I do not think that the argument that existing proof checking work is applicable to only smaller networks is compelling, as (unless I am missing something) proof checking should in practice be faster than verification, and verification scales to much larger models than ACASXu.
> >
> > - Because of being able to capture complex dependencies between piecewise-linear components, MILP solvers are often efficient in verification queries for which non-MILP solvers can struggle. See, e.g., discussion and experiments in [Zhang et al., General Cutting Planes for Bound-Propagation-Based Neural Network Verification] on solvers such as Venus. If the end-goal is the sound verification of a model, then a comparison between (i) MILP-based verification using SoA tools and (ii) non-MILP-based verification + proof checking strengthens the evaluation of the contribution in my view. This is because the former precisely achieves the verification of the verifier in one go. If the overall goal is not the sound verification of the model, then I would agree that such a comparison is not necessary, but the paper does not motivate other practical aims.
> >
> > **Concern on the correctness of the MILP-compilation.** The authors give compelling arguments pertaining to a reduction of the trusted code base. My concern mainly concerns floating-point errors and input splitting however.
> >
> > - Floating point errors. The MILP formulation in Equation (2) uses bounds to encode the ReLUs. From the rebuttal, I understand that the proposed approach uses MILP to optimise all of  the bounds for the inputs to the ReLUs, which alleviates my concern on the soundness of the encoding. I would expect this to fail to scale to big models with big ReLU counts however, also in line with previous work; e.g., [Singh et al., Boosting Robustness Certification of Neural Networks, ICLR19] uses MILP to optimise only some of the ReLUs because of computational requirements. Could the authors further comment on this?
> >
> > - Input splitting. I think that branching heuristics such as BaBSR, which are formulated with respect to symbolic bounds in the bound propagation phase are able to treat input splitting and ReLU splitting uniformly. In practice, even for high-dimensional problems the heuristics may choose to split on an input rather than on a ReLU. So, even if I agree with the authors that the proposed method can in principle be extended to account for input splitting, the paper should make clear how the evaluation dealt with potential input splitting used by the verifiers.
> >
> > **Concern on unsound results.** Given VNNCOMP results where verifiers are believed to produce unsound results (when compared to the results of the majority of the verifiers), and the existence of benchmarks such as  SoundnessBench where the ground truth on the verification result is known [Zhou et al., Testing Neural Network Verifiers: A Soundness Benchmark with Hidden Counterexamples], I believe that the consideration of these verification queries and the validation of the results produced by the proposed approach would strengthen its significance.

---

> ### Author Response · Authors · 2025-08-06
>
> Thanks for your questions. We provide our responses below.
> 1.  MNISTFC_GDVB benchmark. Evaluation on diverse benchmarks.
>
> MNISTFC_GDVB was introduced in [Harnessing neuron stability to improve DNN verification - Duong et. al.] in which they used the GDVB tool [Systematic Generation of Diverse Benchmarks for DNN Verification - Xu et. al.] to create diverse (i) networks, by varying network depth and layer width from a given seed network, and (ii) properties, by perturbing radii values (e.g., 0.01 to 0.09). In particular, we use MNISTFC_GDVB networks ranging from 2-6 layers, 16-512 neurons each layer, and 64-3072 ReLU counts (see Tab.1).  We believe that these networks, along with the others, are sufficiently diverse.
>
> 2. Proof checking is faster than verification?
>
> Proof checking is a lot *slower* than verification:  This might appear counterintuitive and so we'll elaborate. Modern verification tools rely on abstraction for efficiency (e.g., abCrown, NeuralSAT, etc employ advanced abstractions for efficiency), in addition to other heuristics and optimizations --- and many of these are verifiers specific, e.g., tool X uses abstraction Y and heuristic Z.  However, these are also what make DNN verification more complex and error-prone. In fact, in our comment to reviewer Pxf9, we mentioned NeuralSAT has soundness bugs due to its optimizations.
>
> Thus, one major design of an independent checker like APTPChecker is that it eschews all these optimizations of verifiers and systematically traverses and checks the proof tree by MILP.  Therefore checking these proofs becomes slower than generating them (which can be buggy and incorrect).
>
>
> Moreover, the other work on DNN proof checking, Marabou checker, is also very slow -- in fact extremely slow as it spent *days* just to check the results of verifying tiny ACAS XU networks that could be verified by Marabou within seconds.  In other words, Marabou checker spent days checking verification results that were obtained in seconds; this support our claim that proof checking is slower than verification.
>
>
> 3. Verification difficulty depends on non-linear components rather than tunable parameters.
>
> We completely agree -- no argument here -- the difficulty of verification depends largely on non-linear components (ReLUs). However, we are unclear about the comment about tunable parameters as we never said that it would affect verification? Perhaps this was because we listed the parameters in Tab 1, and mentioned that our networks have millions of parameters to indicate that they are not toy models? We will clarify in our writing if that's where the confusion stems from, e.g., we can also include that these models have thousands of ReLUs, which imply high complexity.
>
>
> 4. The goal is not sound verification. Motivate other practical aims.
>
> Indeed, our goal is not on sound verification of the model but instead to verify results of verification tools.  As suggested by reviewer Pxf9, we agree that a concrete example of our tool detecting unsoundness bugs would be helpful. To address this concern during the rebuttal period, we have found a real problem causing unsoundness in NeuralSAT and we will use it to motivate our work.
>
>
> 5. APTPchecker handles big models. Comparison to verification work RefineZono.
>
> After looking at that paper we believe the reason is because we perform bound optimizations (neuron stabilization) at *each* layer.  Our evaluation in sect 5.1 (figs 4 and 5) shows that this significantly reduces the number of constraints and helps computation. In contrast, the ICLR19 paper relies on zonotopes to compute bounds and uses MILP to tighten bounds--but this is *not* performed at each layer and instead at some layer K, for some neurons P.  Note that hardware advances can also make a difference (e.g., we use an AMD Threadripper 64-core vs. Intel Xeon 14-core/i9 10-core).
>
>
> 6. Input splitting for high-dimensional problems. APTP handles input splitting strategy.
>
> We agree that high-dimensional problems might be beneficial from input splitting and will clarify how APTP handles input splitting strategy as mentioned in our rebuttal, i.e., converting each formula encoding hidden activation to input intervals (supported by APTP), to capture the subspace being checked.
>
>
> 7. Example of unsoundness detected by APTPchecker.
>
> Thank you for the suggested paper, which presents both (i) real bugs and (ii) synthetic bugs.  Looking closely at NeuralSAT issue #8 that we mentioned above and to reviewer Pxf9, looks like the reported bug belongs to the category (i) (and from the author of this paper). We will include this example to demonstrate APTPchecker.
> In addition, we have looked at the synthetic bugs, and these injected bugs caused the verifiers to randomly miss branches, which APTPchecker will catch (raise errors as invalid proof) because that is the first thing to check that all branches are visited.  Thanks for mentioning this work and we will include these to motivate our work.

---

> > ### Comment · Reviewer_my7d · 2025-08-08
> >
> > Thank you for the response. My primary concern regarding the evaluation of the method remains unresolved. The rebuttal does not convincingly demonstrate that MILP-based bound optimisations can scale effectively to large networks. In contrast to BaB-based  neural network verifiers, MILP is well-known to struggle with scalability due to the lack of GPU acceleration. Moreover, the argument that proof checking is harder than verification adds further ambiguity to the scalability of the proposed approach. So, while I appreciate the benchmarks considered, both the paper and the rebuttal leave unclear the range and scale of models that the proposed approach can support.
> >
> > The rebuttal did not clarify how input splitting from the verifiers was handled in the evaluation, which further limits the interpretability of the reported results.
> >
> > I appreciate the inclusion of new results on NeuralSAT, which strengthen the empirical evaluation of the approach.
> >
> > After careful consideration, and in light of some concerns being partially addressed, I have increased my score. However, given that my main concern remains, I still believe that the current version of the work is not ready for publication.
> >
> > ***Clarification.*** You used the argument “our benchmark consists of 400 problems across DNNs with up to 1.7M parameters” to explain that the models considered in the evaluation are not toy models.

---

> > > ### Author Response · Authors · 2025-08-09
> > >
> > > Thanks for the comment and the score increment. As there were no technical questions or issues requiring clarification, we will focus more on your thoughts about the designs of APTP.
> > >
> > > 1. MILP does not scale.
> > >
> > > Using MILP is a philosophical choice and may be hard to sway opinion. Indeed, the MILP solver Gurobi that APTP checker and other DNN analyzers rely on, does not support GPU at this moment. However, in the future it might or some other MILP solver might support it, and since APTPchecker uses Guorobi as blackbox, it could immediately benefit from such advancement.
> > >
> > > Second, we are genuinely curious on what the reviewer considers the correct approach for checking generated proof. We use MILP simply because it is the de-facto standard in representing DNN constraints and solving them. Unlike verification algorithms that employ various specialized abstractions and optimizations for efficiency (as you have noted), APTP explicitly avoids such optimizations to serve as a reliable and independent checker. As long as we keep these goals, we are very open to exploring non-MILP approaches.
> > >
> > > Finally, APTP is one of the very first work on proving the results of verifiers. While using MILP might not be the most scalable approach (though as mentioned, arguable and we are open to new ideas), it is the most general and promising approach so far (the existing Marabou checker is specifically for the Marabou verifier and limited to very small ACAS problems; in contrast APTP covers the much broader class of BaB verifiers, provide standard proof format and even an independent checker prototype, and has been eval on proofs on much larger DNN problems).
> > >
> > > The field of DNN verification is in dire need of trust and assurance of verifier results. It needs a starting point, and we believe that APTP provides exactly that---a standard and generizable foundation that future work can build upon.
> > >
> > >
> > > 2. Input splitting
> > >
> > > As we mentioned, this is easy for us and we have already described the technical details in the rebuttal.  We just didn't have enough time during the rebuttal period to implement and evaluate it (we would have prioritized it had this been raised in an explicit question).  We didn't consider it in our paper because while it is a useful optimization to deal with low-dimensional networks, we find it far less interesting and impactful compared to neuron splitting BnB algorithms, which are often key for solving challenging DNN problems and the main focus for improving major DNN verification techniques. That said, this is *easy* for us to do; if additional results are allowed and we'd be happy to do it.

---

### Note · Authors · 2025-08-13

We want to once again thank the reviewers for their time and constructive feedback---both critical and positive. Through the rebuttal and discussion, we were able to perform additional experiments to provide more evidence supporting the work or answering reviewers' questions. We believe we have addressed all main questions and confusions, and appreciate that the reviewers have taken our responses into their consideration and raising their scores accordingly.  While there might still be few reservations (e.g., in using a different approach like MILP vs non-MILP), we hope the reviewers find the changes and additional experiments satisfactory and, like us, are convinced that the work is a significant step toward AI safety and trustworthiness.

---

### Decision · Program_Chairs · 2025-09-17

**Decision:**

Accept (poster)

**Comment:**

The paper focuses on checking the correctness of neural network verifiers by proposing an automatic proof-checking procedure. In particular, it focuses on the branch-and-bound-based neural network verification algorithms, where an activation pattern tree can be generated to represent the subproblems solved during verification. For each unsatisfiable subproblem, a MILP solver is used to verify the proofs. The approach is demonstrated on two neural network verifiers, alpha,beta-CROWN and neuralsat, on various benchmarks from VNN-COMP. The method is relatively simple and straightforward, but as one of the first paper exploring this important topic and the engineering efforts the authors made to enable this framework, I think this paper is worth publishing.

One important concern raised by the reviewer is the scalability of the MILP-based approach. This is indeed a valid concern, but I think it is ok to leave these as an open challenge for future exploration. Additionally, the potential numerical and precision issues are worth discussing.